# Interplay of human macrophage response and natural resistance of infection by *L. (V.) panamensis* to pentavalent antimony

Olga Lucía Fernández[1,2]*, Ashton Trey Belew[3,4], Mariana Rosales-Chilama[1,2], Andrea Sánchez-Hidalgo[1], María Colmenares[5], Nancy Gore Saravia[1,2], Najib M. El-Sayed [3,4,6]*

**1** Centro Internacional de Entrenamiento e Investigaciones Médicas (CIDEIM), Cali, Colombia, **2** Universidad Icesi, Cali, Colombia, **3** Department of Cell Biology and Molecular Genetics, University of Maryland, College Park, Maryland, United States of America, **4** Center for Bioinformatics and Computational Biology, University of Maryland, College Park, Maryland, United States of America, **5** Centro de Investigaciones Biológicas, Consejo Superior de Investigaciones Científicas (CSIC), Madrid, Spain, **6** University of Maryland Institute for Health Computing, North Bethesda, Maryland, United States of America

* elsayed@umd.edu (NME); ofernandez@cideim.org.co (OLF)

## Abstract

Macrophages are the principal host cells of *Leishmania spp.* in human infection and play a critical role in controlling infection and enabling parasite survival and persistence. Nevertheless, understanding of drug resistance in leishmaniasis has primarily focused on the parasite. This investigation provides evidence of the significant differential macrophage response to *ex vivo* infection with clinical strains of *L. (V.) panamensis* naturally resistant (zymodeme 2.3/zym 2.3) or sensitive (zymodeme 2.2/zym 2.2) to antimonial drug, and the distinct effect of this drug on the activation of macrophages. Transcriptome analysis of infected monocyte-derived macrophages from healthy donors revealed significant interferon and cytokine signaling in response to zym 2.3 strains compared to zym 2.2 strains. Furthermore, in the presence of antimony, macrophages infected with zym 2.3 strains, but not with zym 2.2 strains, significantly increased the expression of genes associated with M-CSF-generated macrophages (M-MØ, anti-inflammatory). Notably, macrophages infected with zym 2.3 strains exhibited elevated expression of genes associated with control of inflammatory and microbicidal response, such as the *IDO1/IL4I1-Kyn-AHR* pathways and superoxide dismutase, and downregulation of transporters like *ABC* and *AQP*, compared to macrophages infected with zym 2.2 strains. Remarkably, the majority of these pathways remained upregulated even in the presence of the strong modulatory effect of antimonial drug. Together, these findings demonstrate that the initial and specific parasite-host interaction influences the *ex vivo* macrophage response to antimony. Identification of key pathways in macrophage responses associated with natural resistance to this antileishmanial, enhances understanding of host-response mechanisms in the outcome of *Leishmania* infection and response to treatment.

**Data availability statement:** RNA-Seq data have been submitted to the Short Read Archive (SRA) at NCBI (BioProject: PRJNA1156057). The comprehensive code used for all analyses reported in the manuscript is available on GitHub: https://github.com/elsayed-lab/mac-rophage_response_lpanamensis as well as on Zenodo under the following DOI: https://doi.org/10.5281/zenodo.16944615.

**Funding:** This study was funded by grants from the United States National Institute of Allergy and Infectious Diseases (NIAID) of the National Institutes of Health (NIH) (https://www.niaid.nih.gov/, award number U19AI129910)(NGS), and the Ministry of Science, Technology, and Innovation of Colombia (https://minciencias.gov.co/, contract no. 848-2019, code no. 222984368586)(NGS). O.L.F. was supported in part by the Global Infectious Disease Research Training Program of the Fogarty International Center, NIH (https://www.fic.nih.gov/Pages/Default.aspx, award number D43TW006589). The content is solely the responsibility of the authors and does not necessarily represent the official views of the NIH. The funders had no role in study design, data collection and analysis, decision to publish, or preparation of the manuscript.

**Competing interests:** The authors have declared that no competing interests exist.

## Author summary

Drug resistance and treatment failure are increasingly recognized in human leishmaniasis. Investigation of resistance has predominantly focused on parasite-mediated mechanisms. This study examines the role of host macrophages in natural resistance to antimonial drug. Our findings reveal distinct responses by macrophages infected with *Leishmania (Viannia) panamensis* strains that are naturally resistant to antimonial drug versus sensitive strains, both in the presence and absence of the drug. Distinctively, resistant parasites induced regulatory pathways that modulate inflammatory responses and alter host cell transporter expression, potentially contributing to parasite survival under antimony exposure. The host cell-parasite interaction in the context of drug resistant intracellular infections presents opportunities for innovative therapeutic strategies targeting host cell responses.

## Introduction

Cutaneous leishmaniasis (CL) is a neglected tropical disease that principally affects poor populations around the world. About 95% of CL cases occur in the Americas, the Mediterranean basin, the Middle East and Central Asia (WHO) [1]. The pentavalent antimonial drug (SbV), meglumine antimoniate, continues to be the standard of care for CL in most endemic regions worldwide; however, treatment failure is recognized as a significant clinical and public health challenge [2–4]. The efficacy of SbV treatment varies by geographic region and the species causing infections [5–7]. The vast majority of cases of CL in Colombia are caused by *Leishmania (Viannia) panamensis* [8–10]. Previous investigations by our group documented a persistent natural disparity in susceptibility to antimonial drug as evaluated by *in vitro* assay, between the two most prevalent zymodemes of *L. (V.) panamensis* in Colombia [11]. Demonstration of the contribution of antimonial drug resistance in treatment failure has been elusive, however, recent studies conducted by our group demonstrated a significantly higher failure rate among patients infected with naturally SbV-resistant *L. (V.) panamensis* zym 2.3 strains, than SbV sensitive zym 2.2 and zym 2.1 strains [12].

Mechanisms of resistance to antimony or other antileishmanial drugs in laboratory-selected strains subjected to sustained exposure to high concentrations of drug, have not been found to reproduce or explain resistance in clinical strains [13,14]. The investigation of resistance mechanisms in clinical strains has been limited and with the exception of a few reports [15–17] neither the acquired nor the natural basis of drug resistance detected in clinical strains by *in vitro* evaluation of susceptibility to antileishmanial drugs has been ascertained. Furthermore, the focus of investigation of resistance mechanisms has primarily been on the parasite, with scant consideration of potentiating mechanisms of parasite survival of drug exposure during interaction with the host cell [17–19].

Macrophages play crucial roles as host, effector and immunoregulatory cells during *Leishmania* infection [20] and can either eliminate or enable parasite survival, depending on specific and complex interactions between the parasite and the host cell [21,22]. Previous investigations have shown that strains of *L. (V.) panamensis* isolated from patients with chronic disease induced significantly higher chemokine gene expression than strains from self-healing cutaneous leishmaniasis [22]. Additionally, genes relevant to transport, accumulation and metabolism of antimonials were differentially modulated by antimony sensitive versus resistant strains of *L. (V.) panamensis* [17]. Distinct subpopulations of clinical strains of *L. (V.) panamensis* identified within previous prospective population-based epidemiologic studies [23] were recently shown *in vitro*, *ex vivo*, and in terms of therapeutic response, to exhibit natural resistance (zym 2.3) or sensitivity (zym 2.2) to pentavalent antimony [12]. In this context, we recently investigated the impact of *L. (V.) panamensis* of zym 2.3 and 2.2 on human neutrophils *ex vivo*. The results demonstrated that infection with zym 2.3 strains significantly altered the gene expression of neutrophils, increasing the expression of detoxification pathways and reducing the production of cytokines [24]. Understanding of the bases of natural drug resistance and the participation of the host cell-parasite interaction, and the effect of anti-leishmanial drugs on this interaction, are critical to the identification of more effective host cell targets and for the development of effective therapeutic strategies.

This investigation was conducted to identify host cell responses elicited by infection with naturally resistant zym 2.3 strains of *L. (V.) panamensis* in human macrophages that potentiate or contribute to the diminished susceptibility to antimonial drug associated with treatment failure. To achieve this, transcriptomic profiles of monocyte-derived macrophages following *ex vivo* infection with *L. (V.) panamensis* strains of zym 2.3 and zym 2.2 in the presence or absence of antimonial drug were compared. Our findings reveal that sensitive and naturally antimony-resistant strains of *L. (V.) panamensis* elicit distinct states of macrophage activation and responses to antimonial exposure. Resistant parasites of zym 2.3 induce regulatory pathways of inflammatory macrophage responses and, in the presence of antimony, promote polarization of macrophages to an anti-inflammatory (M-M∅) profile. The results of this investigation contribute to the identification of targetable host cell responses for innovative treatments having potential to mitigate the clinical consequences of natural antimony resistance.

## Methods

### Ethics statement

The study was approved and monitored by the CIDEIM Institutional Review Board (Comité Institucional de Ética de Investigación en Humanos-CIEH) for research involving human subjects in accordance with national and international guidelines for Good Clinical Practice. Every participant provided voluntary, informed, and signed consent. The institutional IRB approval number is 1274.

### Study population

Transcriptome analysis and ROS production were performed using MDMs obtained from nine healthy volunteers, both male (n = 5) and female (n = 4), between 18 and 60 of age, with no clinical history of leishmaniasis. Volunteers resided in the municipality of Cali, Colombia, Recruitment was conducted by the clinical support team of the Centro Internacional de Entrenamiento e Investigaciones Médicas (CIDEIM) in Cali. A blood sample of 200 mL was collected from each volunteer.

### Clinical strains

Twelve *Leishmania (Viannia) panamensis* strains isolated at the time of diagnosis from patients with cutaneous lesions by medical personnel in CIDEIM, and cryopreserved in liquid nitrogen in the institutional biobank were included in this study. Six strains belonging to the zym 2.2 and six strains of zym 2.3 that had been previously evaluated for antimonial drug susceptibility, and species confirmed by serodeme and zymodeme [8,25] were used for *ex vivo* assays within four

passages of recovery from liquid nitrogen. Species identification was achieved by isoenzyme electrophoresis [25] and indirect immunofluorescence using species-specific monoclonal antibodies as described elsewhere [8]. Drug susceptibility of intracellular amastigotes was estimated by evaluation of percentage of parasite survival in PMA-differentiated U-937 cells (ATCC CRL-1593.2) after exposure to pentavalent antimony (SbV) at a final concentration of 32 μg/mL, compared to control without drug exposure [26].

## Culture of monocyte–derived macrophages, infection, and drug exposure

Monocytes were isolated from peripheral blood samples (200 mL) obtained from healthy donors by centrifugation over a Ficoll-Hypaque gradient (Sigma-Aldrich, 10771; USA) followed by purification with magnetic anti-CD14 microbeads (Miltenyi Biotec, LS Columns 130-042-401; USA), according to the manufacturer's instructions. Macrophage cell cultures were generated by adherence of monocytes (>95% CD14 + cells) to cell culture plasticware in a 6-well plates (1 x $10^6$ cells in 3 ml of medium per well) in serum-free RPMI 1640 medium (Gibco, 22400; USA) for 2 h, followed by culture for 7 days in RPMI 1640 media supplemented with 20% heat inactivated FBS (Gibco, 10082; USA) at 37°C in 5% $CO_2$. Promastigotes were cultured in Senekjie medium with a PBS overlay at 25°C for 6 days until reaching stationary phase. Prior to infection, parasites were opsonized during 1 h with 10% AB positive human serum (Sigma-Aldrich H3667; USA). Macrophages were exposed to the parasites for two hours at a ratio of 5 parasites per cell, washed twice with phosphate-buffered saline (PBS) to remove free parasites, and then incubated for 24 hours at 34°C in 5% $CO_2$ using RPMI 1640 medium (R6504; Sigma) supplemented with 10% heat-inactivated fetal bovine serum (FBS; 10082; Gibco). Supernatants were replaced with complete RPMI containing additive-free meglumine antimoniate (Walter Reed 214975AK; lot no. BLO918690–278-1A1W601; antimony analysis, 25% to 26.5%, by weight) at a final concentration of 32 μg SbV/ml for 72 hours as previously described [26]. The concentration of 32 μg SbV/ml, corresponds to the estimated maximum plasma concentration (plasma $C_{max}$) during standard-of-care-treatment [27]. Infected and uninfected macrophages without antimony treatment were incubated for 96 hours in all assays.

## Transcriptomic profiling and data analysis of differentially expressed genes

Macrophages cultured under the described experimental conditions were collected using Trizol, treated with DNAseI, and purified using the RNeasy mini kit (Qiagen, 74704; USA). RNA integrity was assessed using an Agilent 2100 bioanalyzer (RIN ≥ 9). Poly(A)-enriched cDNA libraries were generated using the Illumina TruSeq sample preparation kits and quality and quantity verified using the bioanalyzer and qPCR (KAPA Biosystems). Using dual-index barcoding methods to reduce index hopping, we multiplexed up to 20 samples per NextSeq 1000 P2 run to ensure sufficient read depth (20M 60 bp paired-end reads) per sample. Sequence quality metrics were assessed using FastQC and Trimmomatic [28] was used to remove adapter sequences and trim when the mean quality score fell below 25. Reads were aligned against the human genome as well as the *L. (V.) panamensis* (TriTrypDb release 36) using hisat2 [29]. The resulting alignments were sorted and indexed via samtools [30] and passed to htseq [31] for generating count tables. Genes with <2 reads per million in n samples where n is the size of the smallest group of replicates [32] were removed excluded from the analysis prior to abundance estimation and subsequent differential expression analyses. All sequences generated were submitted to the Short Read Archive (SRA) at the NCBI (BioProject: PRJNA1156057).

## Reactive Oxygen Species (ROS) production

ROS production was measured using a luminol-based chemiluminescence assay, as previously described [33]. Monocytes from five healthy donors were resuspended in an RPMI medium at 5 x $10^5$ c/ml and distributed into the wells (100 μL) of white opaque 96-well microplates (Perkin Elmer, PKE_6005299). Macrophage cell cultures were generated by adherence of monocytes to cell culture plasticware as previously described [17]. Cells were cultured in HBSS

(Sigma-Aldrich, H668; USA) infected with clinical strains of *L. (V.) panamensis* at a ratio of 5 parasites per macrophage, and ROS production was evaluated immediately for one hour. Luminol Sodium Salt (Carbosynth Limited) was added prior to infection at a final concentration of 20 µg/ml, and luminescence induced by ROS was measured every 2.5 min over 60 min using a plate reader (Chamaleon V Multilabel Microplater Reader; Hidex, Finland). The positive control was the induction of ROS by 100 ng/ml PMA.

## Statistical analysis

Biological replicates and batch effects were assessed and visualized using the hpgltools (https://github.com/elsayedlab/hpgltools) R package. The process included creating density plots, boxplots of depth, coefficient of variance, hierarchical clustering analyses, variance partition analyses [34], and PCA before and after normalization. Combinations of normalization and batch adjustment strategies were evaluated. The normalization methods that were typically tested included trimmed median of M-values, relative log expression, and quantile. These were combined with batch evaluation strategies from the surrogate variable analysis package (sva) [35]. Differential expression (DE) analyses were performed using a single pipeline which conducted all pairwise comparisons using the Bioconductor packages: limma [36], edgeR [37], DESeq2 [38], and EBSeq [39]. In each case (except EBSeq), the surrogate variable estimates provided by sva were used to adjust the statistical model in an attempt to address the batch/surrogate effects. The quality of each comparison was evaluated by the degree of agreement among methods, but the interpretations were primarily informed by the DESeq2 results. Pairwise contrasts in genes with a Benjamini-Hochberg multiple-testing adjusted $P$ value of <= 0.05 were defined as differentially expressed. Genes with significant changes in abundance (false discovery rate adjusted P values ≤ 0.05) were passed to gProfileR2 [40], KEGG pathway analyses using ConsensusPathDB [41], and gene set variation analysis (GSVA) [42]. Gene ontology analyses were supplemented with manual data curation. The Mann–Whitney U test was used to establish statistical differences between ROS production induced in macrophages by infection with 2.2–sensitive and 2.3-resistant strains. The comprehensive code used for all analyses reported in the manuscript is available on GitHub: https://github.com/elsayed-lab/macrophage_response_lpanamensis as well as on Zenodo under the following DOI: https://doi.org/10.5281/zenodo.16944615.

## Results

Our research strategy exploited the uniquely disparate susceptibility to antimonial drug of the genetically and phenotypically distinct zym 2.2 and 2.3 subpopulations of *L. (V.) panamensis,* to decipher the potential participation of host macrophages in natural resistance to antimony. To accomplish this, we conducted a comprehensive transcriptome analysis using RNA-seq, of the modulation of pathways in the response of monocyte-derived macrophages (MDMs) from healthy donors to parasite infection and exposure to pentavalent antimony. (Fig 1A).

The responses to a primary set of six clinical strains of *L. (V.) panamensis*, three naturally resistant (zym 2.3) and three sensitive (zym 2.2) to antimony, were assessed in MDMs obtained from three healthy donors. This initial design took into consideration donor-related variability. To determine the robustness and parasite-dependent reproducibility of the transcriptomic expression profiles, an additional, and distinct set of six clinical strains (three zym 2.2 and three zym 2.3) was evaluated in MDMs of another healthy donor. The significant differences observed in the transcriptome profiles across the four donors (encompassing both primary and additional sets of strains) are robust. The consistency of transcriptomic responses within the primary set of strains, visualized through a principal component analysis (**Fig 1B**) and further validated by the additional set, justified the inclusion of all data in the final RNA-seq analysis. Macrophage ROS production in response to infection with zym 2.2 and zym 2.3 strains and drug exposure was evaluated in MDMs from five healthy donors as a potential determinant of intracellular parasite survival in the presence of SbV based on the identified activation pathways.

PLOS Neglected Tropical Diseases

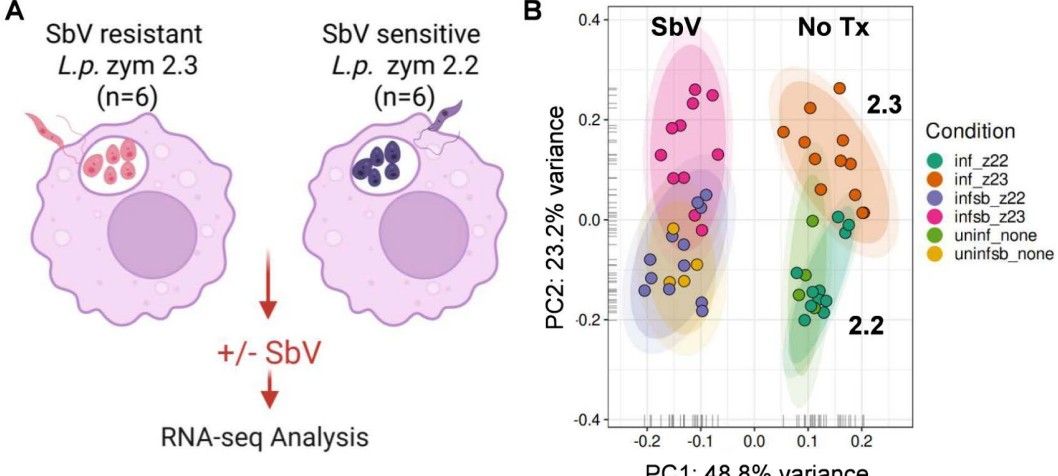

**Fig 1. Global transcriptome profiling of human MDMs infected with zym 2.2 and zym 2.3 of *L. (V.) panamensis* in absence or presence of antimony.** (A) Schematic of experimental strategy (B) Principal Component Analysis (PCA) plot of normalized and SVA-adjusted RNA-seq expression values of macrophage samples from four donors is shown. For all experimental conditions (infected and uninfected), one replicate was used per condition (defined by strain and donor for infected samples, and by donor for uninfected controls). Specifically, MDMs from three donors were challenged with one set of zym 2.2 (n = 3) and zym 2.3 (n = 3) strains, while a fourth donor was infected with a distinct set of zym 2.2 (n = 3) and zym 2.3 (n = 3) strains. Samples represent uninfected controls (none) or infected cells treated with or without 32 µg SbV/mL. Metadata and accessions for all samples used in this study are included in S1 Table. One of the 55 samples (TMRC30162) was excluded from downstream analyses due to the low depth of reads mapping to coding genes.

### Distinct human macrophage activation profiles were induced by antimony exposure and by infection with natural antimony resistant or sensitive strains of *L. (V.) panamensis*

Analysis of global gene expression in human MDMs infected with *L. (V.) panamensis* using principal component analysis (PCA) showed the most pronounced separation (along PC1) between macrophages exposed to antimony versus those that were untreated (Fig 1B), regardless of infection status. Also evident was the separation (along PC2) of macrophages infected with strains of zym 2.3 from those infected with zym 2.2 strains. Remarkably, the same PCA revealed that the gene expression profiles of macrophages infected with the zym 2.2 of *L. (V.) panamensis* clustered together with those of uninfected cells both in the absence or presence of antimony.

The effect of individual donors was a significant source of variance. When the analysis incorporated a model including donor, zymodeme of infecting parasite, and drug, the majority of variance in the data could not be explained by any of these factors. However, significantly greater variance was attributed to donors than to any other factor (S1A Fig). In contrast, when PCA was performed with donor as the primary factor, the result suggested that drug treatment was the dominant factor in the data, followed by zymodeme of the infecting strain (S1B Fig).

### Naturally antimony-resistant strains of *L. (V.) panamensis* elicit a pronounced inflammatory response in conjunction with regulatory pathways

Differential expression (DE) analyses comparing infected and uninfected macrophages revealed that infection by zym 2.3 strains induced a more profound perturbation of the MDM transcriptome than zym 2.2 strains, modulating 646 genes vs. 268 genes, respectively, that were specific to infection by each zymodeme ($\log_2$FC ≥ 1 or ≤ -1; $P_{adj}$ < 0.05) (Fig 2D). Remarkably, the transcriptomic responses of MDMs to infection with zym 2.3 vs. zym 2.2 strains were distinct and quite unique, sharing only 88 up- or downregulated genes. Nine genes exhibited opposing expression patterns in macrophages

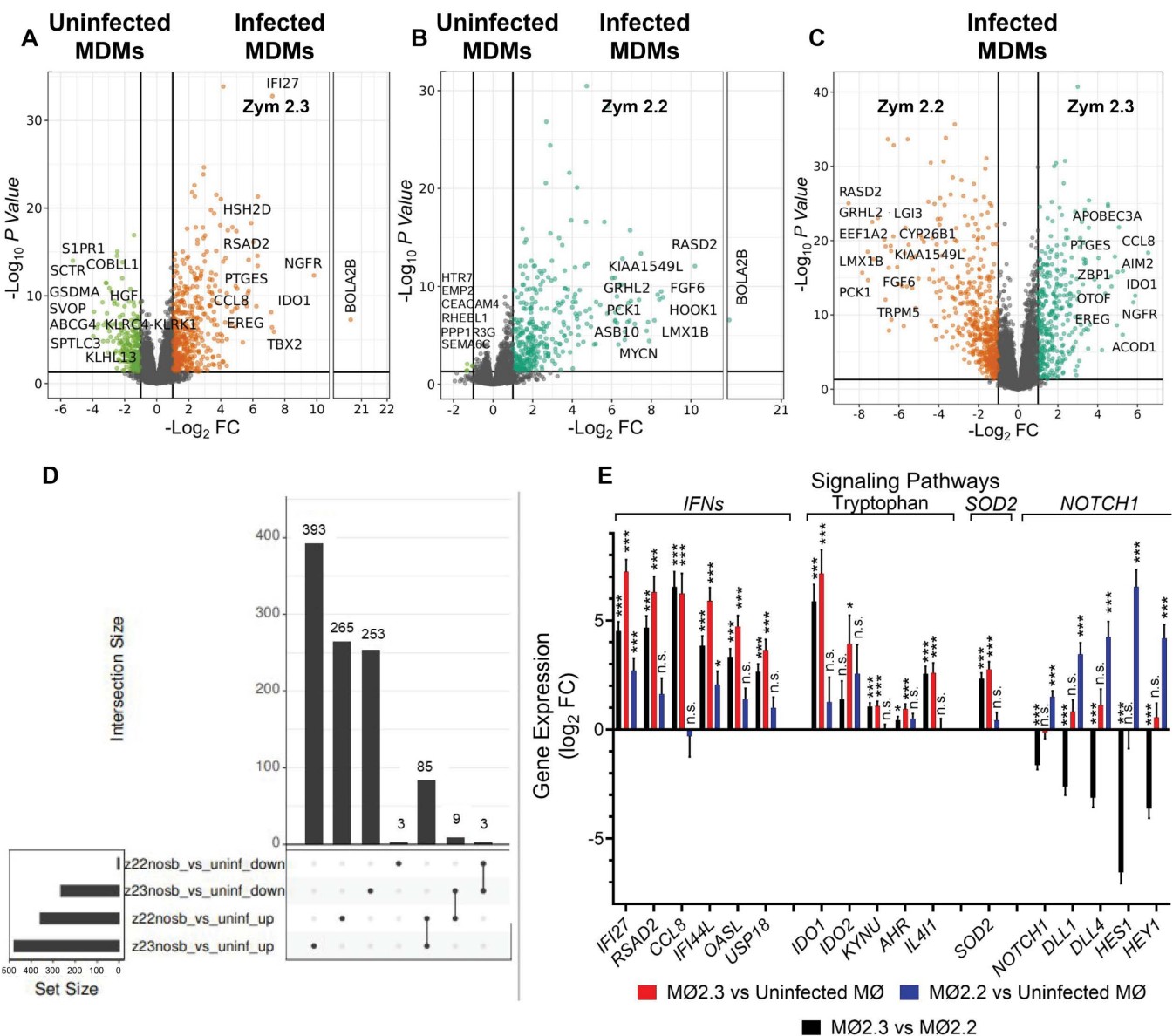

**Fig 2. Differentially expressed genes in macrophages infected with *L. (V.) panamensis* zym 2.2 or zym 2.3.** Volcano plots visualizing comparisons of (A) MDMs infected with zym 2.3 strains vs. uninfected cells; B. MDMs infected with zym 2.2 strains vs. uninfected cells; C. MDMs infected with zym 2.3 strains vs. MDMs infected with zym 2.2 strains. The top 10 up or downregulated genes are labeled in panels **(A-C)** when space allows, and the color scheme is analogous to the one used in Fig 1. (D) UpSet plot summarizing the numbers of unique and shared upregulated and downregulated genes by infection with zym 2.3 and zym 2.2 strains. (E) Differential expression of highlighted genes from selected pathways in macrophages infected with zym 2.3 strains and zym 2.2 strains of *L. (V.) panamensis.* Data corresponds to transcriptome analysis of monocyte-derived macrophages from four healthy donors. Panels A to D: Genes were deemed up- or down-regulated when $\log_2$ FC > 1 or < -1, and adj. $P$ < 0.05. * $P$ < 0.05 ** $P$ ≤ 0.01, *** $P$ ≤ 0.001.

infected with zym 2.3 and zym 2.2 strains (*ABCB5, RFX4, CA14, EGR1, MCF2L, DNASE1L3, FOS, IFITM10, and PKD1L3*) being downregulated in zym 2.3-infected macrophages and upregulated in zym 2.2-infected macrophages. This set of genes encompasses a wide range of cellular functions, including ATP-dependent transmembrane transport, transcriptional regulation, cell proliferation and differentiation. Some of the genes such as *ABCB5* are associated with

skin-resident mesenchymal stem cells that exhibit potent immunomodulatory and wound healing-promoting capacities and antimony transport [43–45], the innate immune response to *Leishmania* (*FOS*) [46], and parasite survival (*EGR1*) [47].

For each comparison carried out, we have illustrated the differential gene expression profiles with volcano plots (Fig 2A-2C), labeling a selection from the significant top 10 up- or downregulated genes. An examination of the lists of differentially expressed genes revealed genes associated with several biological processes, highlighting upregulated genes associated with inflammatory response (*PTGES, CCL8 and EREG*), in the comparison between macrophages infected with zym 2.3 strains and uninfected cells (Fig 2A), as well as in the comparison between macrophages infected zym 2.3 strains and those infected with zym 2.2 strains (Fig 2C). Most remarkably, the expression of *IDO1* (indoleamine 2,3-dioxygenase 1) ranked among the top ten genes that were significantly induced by infection with zym 2.3 strains (Fig 2A and 2C). IDO is a key enzyme in tryptophan catabolism that depletes this amino acid and generates kynurenines, creating an immunosuppressive microenvironment and dampening macrophage proinflammatory responses.

Pathway enrichment analyses based on a set of significantly upregulated genes ($\log_2 FC \geq 1$; $P_{adj} < 0.05$) in macrophages, showed that *L. (V.) panamensis* strains of zym 2.3 induce more activation and regulation of immune response compared to both uninfected and zym 2.2-infected macrophages (S3 Table,). This inflammatory immune response is evidenced particularly by enrichment of pathways, such as "cytokine signaling in the immune system," "Interferon alpha/beta signaling," "interferon gamma signaling," and "chemokine receptors bind chemokines" (in all cases, $P < 0.0001$) (S3 Table). Interestingly, pathways associated with the negative regulation of inflammatory macrophage responses are also upregulated by infection with zym 2.3. These include the tryptophan catabolic process to kynurenine ($P = 0.002$), Interleukin-10 signaling ($P < 0.0001$), and negative regulators of DDX58/IFIH1 signaling ($P = 0.004$), which are key components of the innate immune response to viruses [48–51](S3 Table).

Macrophages infected with zym 2.2 strains, compared with uninfected cells, upregulated a unique set of pathways that distinctly differed from those observed in zym 2.3 infection. These pathways are primarily related to developmental signaling and cellular differentiation, with a particular influence on NOTCH1 pathway: "signaling by NOTCH1 t(7;9) (NOTCH1:M1580_K2555) translocation mutant", $P < 0.0001$. (S3 Table). This repertoire of pathways presents a striking contrast to the immune response of macrophages observed during zym 2.3 infection.

In macrophages infected with zym 2.3 strains, key genes associated with the inflammatory response and its regulation (*AHR, IDO1*, and *IL4I1*) [52,53] were significantly overexpressed in zym 2.3 infection vs. zym 2.2 infection and zym 2.3 infection vs. uninfected macrophages, but not in the zym 2.2 infection vs. uninfected (Fig 2E). This suggests that the infection simultaneously activates both pro-inflammatory and regulatory mechanisms that influence macrophage function. Additionally, genes associated with NOTCH1 were significantly overexpressed in the zym 2.2 infection vs. uninfected cells and downregulated in the contrast of zym 2.3 infection vs zym 2.2 infection (Fig 2E).

Given the crucial role of oxidative stress in the microbicidal activity of macrophages against intracellular pathogens like *Leishmania* [54], and the fact that oxidative stress is a known mechanism of action for antimony [55], we specifically evaluated genes associated with the antioxidative stress response. Comparison of macrophages infected with zym 2.3 strains to uninfected cells revealed a significant upregulation of *SOD2* (~2.8-fold change, $P < 0.001$) (Fig 2E). In contrast, infection with zym 2.2 strains did not induce a significant expression of *SOD2* when compared to uninfected cells (~0.4-fold change, $P = 0.574$). Other important molecules involved in antioxidative stress, such as catalase, glutathione peroxidase, and peroxiredoxins, were also analyzed. However, under the conditions of our assays, they were not significantly modulated.

The pathway enrichment analysis based on downregulated genes revealed distinct patterns for zym 2.3 and zym 2.2 infections of macrophages. In the contrasts between macrophages infected with zym 2.3 and uninfected cells, a subset of the downregulated genes were associated with ABC transporter activity ("ABC-type xenobiotic transporter activity" $P = 0.006$), endocytosis, phagocytosis, and intracellular trafficking (cargo receptor activity, $P = 0.037$; immunoglobulin

receptor activity, *P* = 0.047), S3 Table. Notably, only six genes were downregulated in the comparison between zym 2.2 infection and uninfected cells. When comparing zym 2.3 to zym 2.2 infections, macrophages infected with zym 2.3 were characterized by downregulation of genes associated with ABC transporters: "xenobiotic transport across blood-brain barrier" *P* = 0.013, "transporter activity". *P* = 0.033, (*ABCB1, ABCB5* and *ABCG4*) and AQP (*AQP2* and *AQP3*) transporters S3 Fig), NOTCH1 signaling, pathways involved in immune responses ("response to interleukin-8" *P* = 0.030; "positive regulation of interleukin-2 production" *P* = 0.015), and cellular differentiation and growth ("positive regulation of skeletal muscle cell differentiation" *P* = 0.014). These findings suggest that zym 2.3 infection has a more profound effect on suppressing various host cell processes compared to infection by zym 2.2 strains. The most significantly downregulated genes in the comparison of 2.3 and 2.2 strain infections, relative to uninfected cells, were ABCB1, ABCG4, ABCB5, ABCA9, ABCC2, AQP2, and AQP3 (S3 Fig).

### Infection with natural antimony-resistant strains of *L. (V.) panamensis* elicites sustained regulatory mechanisms of the macrophage inflammatory response despite significant modulation induced by antimonial drug

The differential gene expression profiles of macrophages infected with *L. (V.) panamensis* of either zymodeme, and subjected to antimony treatment, revealed a predominant drug effect that was independent of the parasite zymodeme (Fig 1B). This drug effect was also manifested by the high number of modulated genes (516 downregulated, 306 upregulated) that were shared by uninfected macrophages and those infected with either zym 2.2 or zym 2.3 strains (Fig 3E). In macrophages infected with the zym 2.2 strains, treatment with antimony resulted in a higher number of uniquely downregulated genes (n = 260), compared to the number of uniquely downregulated genes (n = 173) observed during zym 2.3 infection. Conversely, exposure to antimony resulted in the upregulation of a greater number of unique genes (n = 148) in macrophages infected with zym 2.3 strains compared to those uniquely upregulated in macrophages infected with zym 2.2 strains (n = 70). The effect of antimony on uninfected macrophages resulted in significant modulation of unique sets of both up (n = 95) and down (n = 66) regulated genes (Fig 3E).

Volcano plots document similarities in the top ten upregulated or downregulated genes in macrophages infected with zym 2.2 and zym 2.3 strains exposed to antimony (Fig 3A and 3B), as well as in uninfected cells exposed to antimony (Fig 3D). A Spearman rank correlation analysis of the top 50 upregulated and top 50 downregulated genes (100 genes in total) of macrophages infected with zym 2.3 and zym 2.2 in the presence of antimony demonstrated a significant positive correlation (r = 0.7922, *P = 0.0001*) substantiating the predominant effect of the drug in the modulation of macrophage response. However, the comparison between macrophages infected with zym 2.3 and zym 2.2 strains in the presence of antimony substantiates the conservation of genes exclusively induced by infection with 2.3. These include several interferon-related genes such as *IFI44L, RSAD2, ISG20, OASL, CCL8 and ZBP1*, associated with macrophage activation, as well as *NGFR, OTOF, SLC38A5,* and the immunoregulatory gene *IDO1* (Fig 3C).

Reactome pathway database and Gene Ontology gene enrichment analysis of upregulated genes in macrophages exposed to antimony revealed that more than 80% of the top ten pathways identified are shared among uninfected and infected cells in the presence of the drug, independently of the zymodeme of the infecting strain (S4 Table). Some of these pathways, all with a significance of *P* < 0.001, are recognized for their potential relevance in antimony metabolism, such as "metallothionein binding to metals", "response to metal ions", and "detoxification of copper ions" [56,57]. Conversely, a similar analysis of the principal pathways enriched in downregulated genes in the presence of antimony revealed few shared pathways between macrophages infected with zym 2.3 and zym 2.2 strains. These shared pathways are principally related to connective tissue disorders [58] (For example, "Defective B4GALT7 causes EDS, progeroid type" *P* = 0.013 for zym 2.3 infection and *P* = 0.05 for 2.2 infection) and eosinophil cell migration ("regulation of eosinophil migration" *P* = 0.039 for zym 2.3, and "eosinophil migration" *P* = 0.01, for zym 2.2). Notably, these pathways were not downregulated in uninfected macrophages exposed to the drug, suggesting that their downregulation is specific to the combined effect of *Leishmania* infection and antimony exposure.

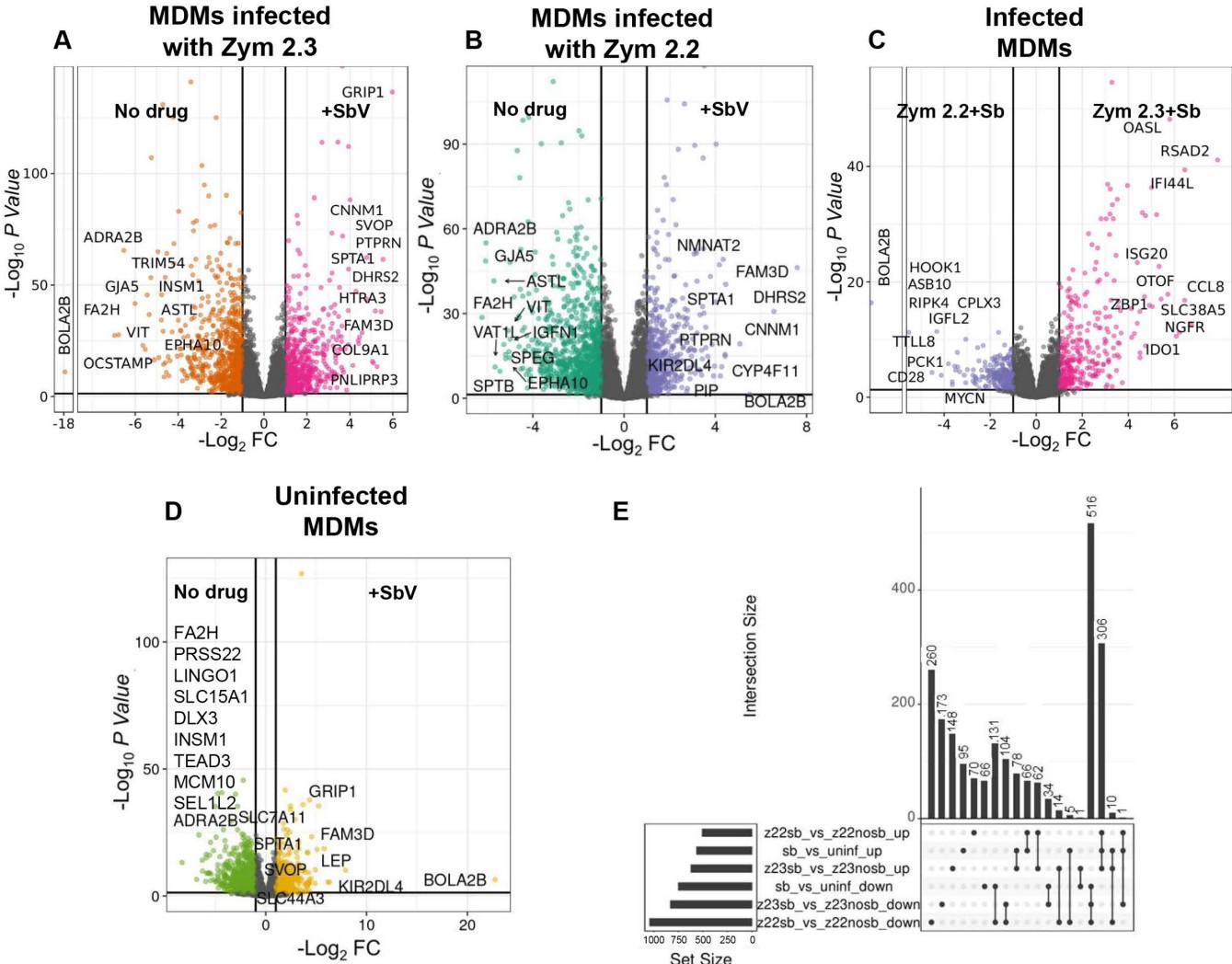

**Fig 3. Differentially expressed genes in macrophages infected with zym 2.2 or zym 2.3 strains, in the presence of SbV as meglumine antimoniate.** Volcano plots visualizing comparisons of: (A) MDMs infected with zym 2.3 strains + SbV vs. MDMs infected with zym 2.3 strains; (B) MDMs infected with zym 2.2 strains + SbV vs. MDMs infected with zym 2.2 strains; (C) MDMs infected with zym 2.3 strains + SbV vs. MDMs infected with zym 2.2 strains + SbV. (D) MDMs uninfected + SbV vs. MDMs uninfected. (E) UpSet plot summarizing the numbers of unique and shared upregulated and downregulated genes by infection with zym 2.3 and zym 2.2 strains, in the presence of SbV. The top 10 up or downregulated genes are labeled in panels (A-D), and the color scheme is analogous to the one used in Fig 1. Data correspond to transcriptome analysis of MDMs from four healthy donors. Number of genes up- or down-regulated with fold change > 1 or <-1, and adj. $P < 0.05$.

Analysis of Reactome pathways based on the comparison of macrophages infected with zym 2.3 vs. zym 2.2 in the presence of antimony showed that infection with zym 2.3 induced a marked enrichment of pathways associated with an interferon response, including "Interferon alpha/beta signaling," $P < 0.0001$ "interferon gamma signaling," $P < 0.0001$, and "chemokine receptors binding chemokines" $P = 0.003$, similar to the response that was elicited by infection in the absence of the antimonial drug (S4 Table). Additionally, pathways involved in telomere packaging ("packaging of telomere ends" $P = 0.007$) and DNA repair processes, such as depurination and purine damage recognition ("depurination" $P = 0.012$, "cleavage of the damaged purine" $P = 0.012$), were also upregulated. Enriched Gene Ontology biological processes further highlighted pathway involved in "regulation of chronic inflammatory response" $P = 0.011$(S4 Table).

Remarkably, the analyses of genes downregulated by infection with zym 2.3 compared to zym 2.2 in the presence of antimony revealed downregulation of pathways associated with "aquaporin-mediated transport" $P = 0.005$ (*AQP2, AQP3, and AQP8*) and ABC transporter transcription activity "transmembrane transporter activity" $P = 0.005$ (*ABCB1 and ABCG4*) S3 Fig. Interestingly, *ABCG4*, *AQP2*, and *AQP8* were upregulated by antimony in uninfected macrophages. These findings are particularly significant as aquaporins and ABC transporters have been previously recognized for their role in antimony transport and drug susceptibility [17,57].

## Treatment with antimony significantly increased the expression of genes associated with M-CSF-generated Macrophages (M-MO, anti-inflammatory) during infection with naturally antimony-resistant strains

Based on the gene sets that define the transcriptome of human monocyte–derived proinflammatory GM-MØ ("Proinflammatory gene set") or anti-inflammatory M-MØ ("Anti-inflammatory gene set") previously reported (GSE68061) [59], we analyzed and identified a notable over-representation of genes associated with the proinflammatory profile of human macrophages (GM-MØ-specific gene set) across infections with both zymodemes (**Fig 4A**). However, a distinct negative enrichment of genes associated with an anti-inflammatory profile (M-MØ-specific gene set) was observed only in infections with the zym 2.3 strains, suggesting a stronger proinflammatory response to these infections compared with zym 2.2 strains. Interestingly, GSEA revealed that antimony modulates the macrophage expression profile in a manner dependent on the infecting zymodeme. In macrophages infected with zym 2.3 strains, treatment with antimony resulted in a negative enrichment of genes associated with a proinflammatory profile and a positive enrichment of genes associated with an anti-inflammatory profile ((M-MØ-specific gene set; GSE68061) (**Fig 4B**). In macrophages infected with zym 2.2, antimony exposure decreased the expression of genes linked to the proinflammatory response without significant changes in genes related to the anti-inflammatory response. Moreover, an evaluation of the effect of antimony on uninfected macrophages indicated a negative enrichment in proinflammatory gene sets, without changes in the expression of anti-inflammatory genes (**Fig 4C**). These findings underscore the nuanced role of antimony in modulating immunoinflammatory responses, to the infecting strain of *Leishmania*.

The positive association of macrophages infected with zym 2.3 strains and the enrichment of genes associated with an anti-inflammatory profile (M-CSF), combined with higher expression of genes associated with regulation of the inflammatory response and microbicidal activity (*IDO1/2*, *IL4I1*, *KYNU*, and *AHR*), are consistent with the significant parasite burden in infections with zym 2.3 compared to zym 2.2 in the presence of antimony. This was supported by the measurement of gene expression corresponding to intracellular amastigotes (S2 Fig).

## Reduced production of ROS concurs with regulatory pathways induced in macrophages infected with naturally SbV-resistant strains of *L. (V.) panamensis*

Oxidative stress is characterized by high levels of reactive oxygen species (ROS), which have direct antimicrobial activity against pathogens and have been reported to be relevant in the microbicidal activity against *Leishmania* infection, depending on the parasite species [54]. Interestingly, the results obtained in the present study show that macrophages infected with zym 2.3 exhibit a higher expression of genes and pathways associated with the regulation of oxidative stress and proinflammatory response, such as *SOD2*, the *IDO1/IL4I1-Kyn-AHR* pathway, and *IL4I1*. Because SOD isoforms constitute the major antioxidant defense systems against ROS, [60] and the tryptophan-degrading reaction catalyzed by IDO is linked to the scavenging of ROS, [61] we evaluated the production of these gene products in infected MDMs from healthy donors. Results demonstrated that human macrophages infected with zym 2.3 of *L. (V.) panamensis* generate significantly lower ROS response compared to macrophages infected with zym 2.2 strains (**Fig 5**). Hence, the regulation of ROS production may be a mechanism that favors the survival of the Sb-resistant zym 2.3 of *L. (V.) panamensis*. It should be noted that the induction of ROS is a reported mechanism of antileishmanial activity for antimony [55].

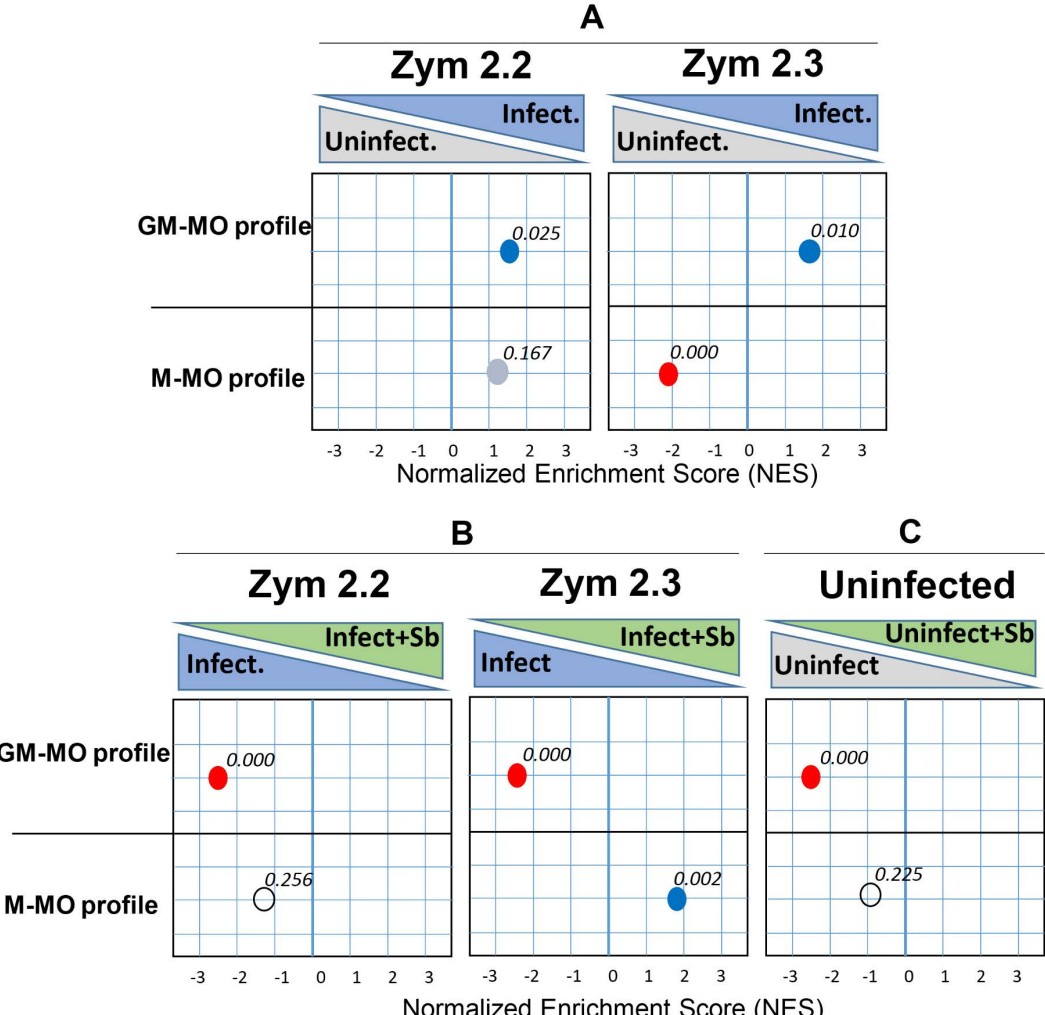

**Fig 4. Gene set enrichment analysis of the global transcriptome profiles of macrophages exposed to different zymodeme infections and treatment conditions.** GSEA on the ranked list of genes obtained from the reported comparison of the transcriptomes of M-MØ+LPS (anti-inflammatory MØ) versus GM-MØ (proinflammatory MØ) + LPS (GSE68061). GSEA of M-MØ- and GM-MØ-specific gene sets on the comparison of the **(A)** macrophage infected with zym 2.3 and zym 2.2 strains vs uninfected, **(B)** macrophage infected with zym 2.3 and zym 2.2 strains in presence of SbV vs infected macrophages, and **(C)** uninfected macrophages plus SbV vs uninfected cells. Normalized Enrichment Score (NES) and False Discovery rate q value (FDRq) are indicated. Scores highlighted in red or blue indicate a significant negative or positive enrichment value for each of the profiles.

## Discussion

This investigation provides a comprehensive transcriptomic analysis of the distinct human macrophage responses to *ex vivo* infection with *Leishmania (Viannia) panamensis* strains that are naturally resistant or sensitive to antimonial drug. Our results reveal different macrophage activation states induced by infection with antimony sensitive (zym 2.2) and resistant (zym 2.3) strains, both in the absence and presence of antimony. Through the elucidation of macrophage responses in the context of natural antimony resistance, we have identified two differences in macrophage activation potentially associated with this resistance: regulation of the inflammatory and microbicidal response, and changes in the expression of host cell transporters. These differences could explain the higher parasite survival of zym 2.3 strains compared to zym 2.2 strains in the presence of antimony in *ex vivo* drug susceptibility assays. Our results generate new hypotheses regarding

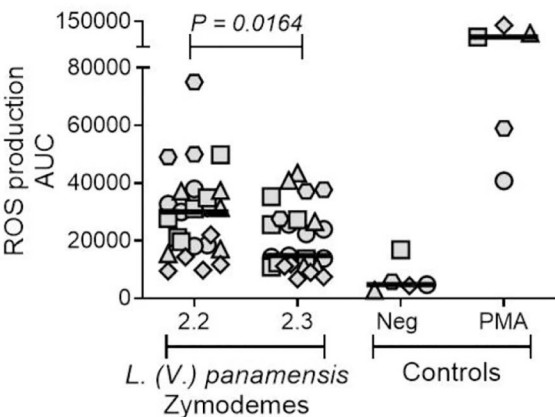

**Fig 5. ROS production by human macrophages infected with clinical strains of _L. (V.)_ panamensis pertaining to zym 2.3 and zym 2.2.** Macrophages infected with zym 2.3 strains of _L. (V.) panamensis_ (n = 6) compared with macrophages infected with zym 2.2 strains (n = 6), MDMs from healthy donors: n = 2 to 5 for each strain. Each geometric form corresponds to an individual donor. Horizontal bar represents median of AUC of ROS production, determined from relative light units (RLU).

the molecular mechanisms of macrophage responses to infection and to antimonial drugs that could contribute to diminished or loss of susceptibility and exemplify the complex adaptive interplay between the parasite, host cell, and antimonial drugs.

Analysis of macrophage responses revealed that antimony-resistant zym 2.3 strains induce a more profound perturbation of gene expression compared to sensitive zym 2.2 strains, modulating more than double the number of genes. Notably, zym 2.3 strains activated pathways in macrophages associated with a stronger inflammatory response than zym 2.2, evidenced by upregulation of interferon signaling, inflammatory cytokines, and chemokines. However, this activation profile was accompanied by induction of regulatory pathways that modulate the inflammatory and microbicidal response, a phenomenon specific to infection by zym 2.3 strains. A key finding was the significant upregulation of _IDO1_ in macrophages infected with zym 2.3. This gene was among the top ten upregulated genes, with a fold change of 7.1 compared to uninfected cells, and 5.9 compared to macrophages infected with zym 2.2 strains. Interestingly, despite the strong effect of antimony on macrophage gene expression, upregulation of _IDO1_ persisted in zym 2.3-infected macrophages exposed to antimony, maintaining a fold change of 6.1 compared to zym 2.2-infected macrophages in the presence of antimony.

IDO1 is an interferon (IFN-I/II) inducible enzyme whose activity can inhibit the growth of susceptible intracellular pathogens by catalyzing the oxidative cleavage of the indole ring of L-tryptophan and depleting pools of this essential amino acid. However, it can also mediate macrophage regulation of a proinflammatory response and contribute to pathogen survival through induction of metabolites, production of regulatory cytokines, and control of oxidative stress [62]. Enrichment analysis revealed overexpression of other IDO pathway-associated genes in zym 2.3-infected macrophages, including _IDO1/2_, and _AHR_. Notably, the _IDO1-Kyn-AHR_ pathway has been described as a survival mechanism for other intracellular pathogens, including _Mycobacterium tuberculosis_ and _Toxoplasma_ [63]. Previous studies have shown that upregulation of IDO1 by _L. donovani_ and _L. major_ can favor parasite persistence in BALB/c mouse macrophages and local lymph nodes, and IDO1 inhibition reduced both local inflammation and parasite burden [64,65]. In contrast, studies in _L. (V.) panamensis_-infected BALB/c mice showed that blocking IDO1 exacerbated the disease, enlarging lesions and increasing the parasite load [66]. However, in lesions from patients with cutaneous leishmaniasis infected with _L. (V.) panamensis_ or _L. (V.) braziliensis_, _IDO1_ expression was downregulated during healing in chronic patients and positively correlated with IFNγ expression [67]. These latter findings align with our results in human macrophages, in which those infected with zym 2.3 strains overexpressed genes involved in interferon response and high expression of _IDO1_.

Interleukin 4-induced gene 1 (*IL4I1*) was also significantly upregulated in macrophages infected with zym 2.3, both in the absence and presence of antimony. IL4I1 belongs to the L-amino-acid oxidase (LAAO) family and catalyzes the oxidation of tryptophan. It contributes to the suppressive activities of macrophages and promotes differentiation of alternatively activated M2 macrophages with anti-inflammatory properties [68]. Similar to *IDO1*, *IL4I1* expression can create a local anti-inflammatory microenvironment. High levels of *IDO1* and *IL4I1* mediate aryl hydrocarbon receptor (*AHR*) activation through tryptophan catabolism. *AHR,* also significantly upregulated by infection by *zym 2.3* strains, is a ligand-activated transcription factor that regulates various biological processes, including immune regulation. Both IDO and IL4I1 have been associated with immunosuppression in tumors through the generation of the *AHR* agonist kynurenic acid (KynA), further underscoring the importance of these molecules in *AHR* signaling. These two enzymes have been proposed as concomitant targets to achieve repression of anti-tumor immunity [53]. The co-expression of *IDO1* and *IL4I1* as activators of *AHR* supports the potential participation of the *IDO1/IL4I1-KYN-AHR* pathways in the downregulation of the antimicrobial response of macrophages infected with zym 2.3 strains.

In concurrence with the overexpression of genes of the *IDO1/IL4I1-KYN-AHR* pathways and its association with the inhibition of microbicidal responses, macrophages infected with zym 2.3 significantly overexpressed superoxide dismutase 2 (*SOD2*) gene compared to both uninfected macrophages and those infected with zym 2.2 strains. SOD isoforms are the major antioxidant defense systems against reactive oxygen species (ROS) [60], one of the principal antimicrobial mechanisms against pathogens [69]. Our functional studies of ROS production corroborated these transcriptomic findings. Macrophages infected with zym 2.3 strains generated significantly lower ROS responses compared to those infected with zym 2.2 strains. Similar results were observed in neutrophils exposed *ex vivo* to infection with zym 2.3 and zym 2.2 strains, whereby zym 2.3 strains induced significantly lower ROS production compared to zym 2.2 strains, both in the presence and absence of SbV [33]. This reduced ROS production in zym 2.3-infected macrophages may represent one of the mechanisms favoring parasite survival, and resistance of infections by zym 2.3 strains to antimonials. Hence, overexpression of *SOD2*, in conjunction with other mechanisms regulating macrophage activation, such as the activation of the *IDO1/IL4I1-KYN-AHR* pathways, could contribute to natural resistance to antimonials in zym 2.3 infections. However, our results showed that the addition of antimonial drug significantly reduced *SOD2* expression in macrophages infected with zym 2.3, compared to infection in the absence of antimonial drug. Functional assays of these pathways are needed to define their precise role in the diminished susceptibility of strains of the zym 2.3 subpopulation of *L. (V.) panamensis* to antimonial drug.

The differences in the activation profiles of macrophages elicited by infection with zym 2.2 and zym 2.3 of *L. (V.) panamensis* not only define the macrophage response to infection but also have implications for the *ex vivo* and significantly associated *in vivo* therapeutic activity of antimony. Remarkably, the addition of antimony to macrophages infected with zym 2.3 significantly increased the expression of genes associated with M-CSF-generated macrophages (M-MØ, anti-inflammatory), as revealed by Gene Set Enrichment Analysis based on ranked comparison of the transcriptomes of M-MØ versus GM-MØ (GSE68061). This shift or polarization towards an anti-inflammatory phenotype (M-MØ) in the presence of antimony could potentially contribute to drug tolerance/resistance by creating a more permissive environment for parasite survival. Similar results have been observed with other intracellular pathogens, such as *Mycobacterium tuberculosis* and *Salmonella typhimurium*. *Mycobacterium tuberculosis* infection can induce both M1 and M2 macrophage polarization, but M2 macrophages are more abundant in chronic infections [70,71]. Treatment of *Mycobacterium tuberculosis*-infected macrophages has been shown to promote M2 polarization, which may contribute to the lower susceptibility to antimicrobial drug [72]. *Salmonella typhimurium* infection can induce both M1 and M2 macrophage polarization [73], but M2 macrophages are more abundant in drug resistance. Overall, these results suggest that the differential modulation of macrophage activation responses reflecting polarization towards an anti-inflammatory profile may be a common feature of intracellular pathogens and this adaptive capacity has important implications for the development of novel, more effective immunomodulatory and antimicrobial host-directed therapies.

In contrast, transcriptome analysis of macrophages infected with zym 2.2 strains highlighted the role of NOTCH signaling 1 in the macrophage response. NOTCH-1 signaling is known to promote M1 polarization through reprogramming of macrophage metabolism, characterized by high production of mitochondrial ROS [74]. This aligns with the pro-inflammatory profile (GM-CSF) of macrophages generated during infection with zym 2.2, as identified by GSE analysis, and concurs with the higher ROS production observed in these macrophages compared to those infected with zym 2.3 strains. On the other hand, addition of antimonial drug significantly downregulated the genes associated with NOTCH 1 signaling, resulting also in a negative enrichment of gene expression associated with a pro-inflammatory profile (GM-CSF). In contrast to the infection with zym 2.3 strains, the addition of antimonial drug to macrophages infected with zym 2.2 did not increase the expression of genes associated with an anti-inflammatory profile (M-CSF). These findings highlight the distinct activation profiles of macrophages infected by zym 2.2 and 2.3 strains, both in the presence and absence of antimony.

Despite the differences induced by antimony exposure in the polarization profile of macrophages infected with zym 2.2 and 2.3 strains, we found that that 80% of the top ten pathways identified were shared among uninfected and infected cells and therefore, associated with the effect of antimony. These pathways are principally related to cellular mechanisms to counteract and prevent metal-induced toxicity, as evidenced by the enrichment of pathways involving metallothionein binding to metals, response to metal ions, and detoxification of copper ions. The participation of metallothioneins in the response to antimony in macrophages infected with *Leishmania* has been previously reported and associated with facilitating Sb-dependent killing of *Leishmania* [56]. These results suggested that the metallothioneins scavenging function could promote Sb accumulation within the phagolysosome of infected macrophages, favoring intracellular killing of *L. (V.) panamensis* [57]. Additionally, it is recognized that metallothioneins play a crucial role in cellular redox balance, supporting their participation in the reduction of the pentavalent antimony pro-drug to the trivalent form having greater antileishmanial activity. These results are consistent with the significant reduction of parasite burden by exposure to antimony during *ex vivo* infection with both zym 2.3 and 2.2 strains. However, independent of the marked effect of antimony on the transcriptome of macrophages, according to the GSE analysis classification, the anti-inflammatory profile of macrophages infected with zym 2.3, could explain the significant intracellular parasite survival of zym 2.3 strains in the presence of antimonial drug.

Interestingly, our analysis revealed distinct patterns in the expression of transporter genes by macrophages during zym 2.3 and zym 2.2 infections. Notably, infection with zym 2.3 in comparison with zym 2.2, in the presence of antimony led to the downregulation of pathways associated with aquaporin-mediated transport (*AQP2, AQP3,* and *AQP8*) and ABC transporter activity (*ABCB1* and *ABCG4*). Given the known role of these transporters in antimony uptake and drug susceptibility, these findings provide insight into potential mechanisms involved in the survival of zym 2.3 strains during exposure to antimonial drugs. The downregulation of aquaporins could potentially reduce antimony influx into the cell, while the reduced expression of ABC transporters might alter the intracellular distribution of the drug. These changes in transporter expression, combined with the anti-inflammatory profile (M-CSF) induced by zym 2.3 strains in the presence of antimony, could create a cellular environment that enables parasite survival despite exposure to leishmanicidal concentrations of antimonial drug. Furthermore, the differential regulation of these transporters in zym 2.3 and zym 2.2 infections underscores the subpopulation-specific adaptations that may contribute to resistance to the antileishmanial effect of antimonial drugs and treatment failure.

## Limitations

This study examined *Leishmania (Viannia) panamensis* strains (zym 2.2 and zym 2.3) and their interaction with human macrophages, yielding insights into natural resistance to antimony observed *ex vivo*. Importantly the clinical response to treatment with antimonial drug of the corresponding patients was significantly associated with the susceptibility to this drug *in vitro* and *ex vivo*. Nevertheless, the use of an *ex vivo* model with human primary macrophages does not fully replicate

the complexity of the tissue microenvironment, as it lacks other immune cells and systemically acting host responses. The absence of T-lymphocytes, dendritic cells, or NK cells in our *in vitro* model does not allow us to evaluate the cell-cell crosstalk and feedback loops that are crucial for modulating macrophage responses *in vivo.* Consequently, this may limit translation of our observations to *in vivo* conditions. Nevertheless, the main goal of this research was to understand the participation of the host macrophage in natural resistance of a subpopulation of *L. (V.) panamensis* to antimonial drug using an *ex vivo* model, which is significantly associated with treatment failure in patients. Extending our findings on the participation of the host cell response to other *Leishmania* species and with different drugs is a long-term goal of ours, as these may vary. In addition, our study focused on transcriptomic analysis, which does not assess protein levels, enzymatic activity, or functional effects, other than ROS production, requiring further proteomic and functional studies. Key pathways (e.g., *IDO1/IL4I1-KYN-AHR*, *SOD2*, and transporter regulation) were identified, highlighting the need for future mechanistic studies such as gene knockouts and pharmacological inhibition. Since the IDO1/IL4I1-KYN-AHR pathway is a target for advanced cancer drugs, these compounds are promising candidates for drug repositioning and should be evaluated in future functional studies. Finally, the single time point analysis may miss the potential dynamic changes in macrophage responses, hence the relevance of longitudinal studies to understand temporal regulation.

## Conclusions

The results of this study reveal that naturally SbV-resistant *L. (V.) panamensis* zym 2.3 strains elicit a unique macrophage response characterized by a strong but regulated inflammatory profile, reduced ROS production, and altered expression of drug transporters. These factors likely contribute to the generation of an intracellular environment favoring parasite survival, thereby participating in resistance to the antileishmanial effect of antimonial drug. Understanding the interplay of complex host-parasite interactions provides valuable insights into the bases, and mechanisms of natural resistance to this widely used anti-leishmanial.

The identification of the IDO1/IL4I1-KYN-AHR pathways as potential mediators of host-cell-mediated resistance provides a crucial target for therapeutic intervention. These findings are of particular significance for drug development, as the same pathways are central to the pathophysiology of other diseases, such as cancer. Consequently, several drugs targeting these specific molecules are already in advanced stages of clinical development. We speculate that these existing drugs could be excellent candidates for drug repositioning, offering a novel co-adjuvant strategy for antileishmanial treatment. This approach could mitigate the clinical consequences of natural resistance by targeting the host-cell mechanisms that promote parasite survival.

## Supporting information

**S1 Fig. Global transcriptome profiling of human MDMs infected with zym 2.2 and zym 2.3 of *L. (V.) panamensis* in absence or presence of antimony**. Principal component analysis (PCA) plot analysis of (A) normalized and SVA-adjusted (for donor) RNA-seq expression values of macrophage samples from 4 donors, uninfected (none) or infected with strains of zym 2.2 and zym 2.3, in the absence or presence of 32 µg SbV/mL, or (B) same analysis without SVA adjustment. Results (including a variance partition plot available on our complete log of analyses at https://doi.org/10.5281/zenodo.16944615) suggest that the majority of the attributable variance is associated with donor over any other factor. In contrast, when PCA was performed with donor as the primary factor, the results indicated that drug treatment was the dominant factor in the data, followed by infecting zymodeme.
(TIF)

**S2 Fig. Parasite burden in macrophages infected by *L. (V.) panamensis* zymodemes 2.2 and 2.3, exposed or unexposed to antimony treatment.** The number of reads mapped to the parasite was observed in every sample, collected by infection status, and plotted on the log2 scale. The small numbers of reads observed in the uninfected samples

shows the low rate of spurious mapping to highly conserved regions; which is similar to the number of reads observed in the drug-treated zymodeme 2.2 samples. The drug-treated zymodeme 2.3 samples retained a significantly higher number of parasite reads, while the untreated samples have the most reads and are indistinguishable from each other. Each red dot describes the mean number of reads observed in that sample group. The box defines the inner-quartile range. The whiskers delineate the outer quartiles. Each colored dot within the violin plot describes the mapping rate of that sample; the density of samples defines the shape of the violin. The degree of significance was defined by a Bonferroni-Holm adjusted $P$ value.
(TIF)

**S3 Fig. Differential expression of highlighted genes from ABC and AQP transporters in macrophages infected with zym 2.3 strains and zym 2.2 strains of** *L. (V.) panamensis.* **(A) Differential gene expression between macrophages infected with 2.3 and 2.2 strains in the absence of drug.** (B) Differential gene expression between macrophages infected with 2.3 and 2.2 strains in the presence of antimony. Data corresponds to transcriptome analysis of monocyte-derived macrophages from four healthy donors. adj. $P < 0.05$. * $P < 0.05$ ** $P \leq 0.01$ ***.
(TIF)

**S1 Table. Samples analyzed and associated metadata.**
(XLSX)

**S2 Table. Differential gene expression analysis.**
(XLSX)

**S3 Table. Top 10 most enriched pathways among up- and downregulated genes in primary human macrophages infected with** *L. (V.) panamensis* **strains of zymodemes 2.2 and 2.3 in the absence of drug.**
(DOCX)

**S4 Table. Top 10 most enriched pathways among up- and downregulated genes in primary human macrophages infected with** *L. (V.) panamensis* **strains of zymodemes 2.2 and 2.3 in the presence of antimony.**
(DOCX)

## Acknowledgments

We are grateful to Diane McMahon-Pratt for her advice during the conduct of the study and review of the resulting manuscript, and Fabienne Tacchini-Cottier for sharing her insight and perspective on the activation of the Notch pathway in the macrophage response to antimony sensitive and resistant subpopulations of *L. (V.) panamensis.* We wish to acknowledge the technical support of Diana Giron in standardizing the laboratory protocols for macrophage infection; Maryury Delgado, Liliana Fernanda López, Mónica Oviedo, and Isabel Guasaquillo for biorepository services and strain characterization; and the CIDEIM clinical research team for their support in the enrollment of healthy volunteers and clinical monitoring of the study. All the sequencing was conducted at the Brain & Behavior Institute - Advanced Genomic Technologies Core (BBI-AGTC) at the University of Maryland, College Park. We would like to acknowledge that figures were created using BioRender (https://www.biorender.com/).

## Author contributions

**Conceptualization:** Olga Lucía Fernández, María Colmenares, Nancy Gore Saravia, Najib El-Sayed.

**Data curation:** Ashton Trey Belew, María Colmenares.

**Formal analysis:** Olga Lucía Fernández, Ashton Trey Belew, María Colmenares, Najib El-Sayed.

**Funding acquisition:** Olga Lucía Fernández, Nancy Gore Saravia, Najib El-Sayed.

**Investigation:** Olga Lucía Fernández, Ashton Trey Belew, Mariana Rosales-Chilama, Andrea Sánchez-Hidalgo, María Colmenares, Nancy Gore Saravia, Najib El-Sayed.

**Methodology:** Olga Lucía Fernández, Mariana Rosales-Chilama, Andrea Sánchez-Hidalgo, María Colmenares, Nancy Gore Saravia, Najib El-Sayed.

**Project administration:** Olga Lucía Fernández.

**Software:** Ashton Trey Belew.

**Supervision:** Nancy Gore Saravia, Najib El-Sayed.

**Validation:** Olga Lucía Fernández, Ashton Trey Belew, Najib El-Sayed.

**Visualization:** Olga Lucía Fernández, Ashton Trey Belew, María Colmenares, Najib El-Sayed.

**Writing – original draft:** Olga Lucía Fernández, Najib El-Sayed.

**Writing – review & editing:** Ashton Trey Belew, Mariana Rosales-Chilama, Andrea Sánchez-Hidalgo, María Colmenares, Nancy Gore Saravia.

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
