## [Decision Letter · Decision Letter 0]

29 Jul 2025

Response to Reviewers
Revised Manuscript with Track Changes
Manuscript

Shaden Kamhawi

co-Editor-in-Chief

Paul Brindley

co-Editor-in-Chief

**Additional Editor Comments:**
**Journal Requirements:**

3) We notice that your supplementary Figures, and Tables are included in the manuscript file. Please remove them and upload them with the file type 'Supporting Information'. Please ensure that each Supporting Information file has a legend listed in the manuscript after the references list.

**Reviewers' comments:**

**Key Review Criteria Required for Acceptance?**

**Methods:**

-Are the objectives of the study clearly articulated with a clear testable hypothesis stated?

-Is the study design appropriate to address the stated objectives?

-Is the population clearly described and appropriate for the hypothesis being tested?

-Is the sample size sufficient to ensure adequate power to address the hypothesis being tested?

-Were correct statistical analysis used to support conclusions?

-Are there concerns about ethical or regulatory requirements being met?

Reviewer #1: - The objectives of the study are clearly presented. The stated objective of the authors was to identify host cell responses specific to antimony-resistant L. (V.) panamensis that could potentially account for reduced drug susceptibility of these parasites.

- The experimental set up is clearly and thoroughly described. The authors take care to ensure scientific robustness at every stage, including ensuring parasites are passage in vitro no more than 4 times (to avoid potential changes while in culture).

- A large sample size is used for the RNA-seq experiments, with 6 different parasite strains (3 from each zymodeme), with 4 different donors. However, greater clarity could be provided as to the total number of experiments performed for each condition (see 'Modifications' section).

- Statistical analyses and workflows are well-designed and robust.

- Ethical approval was obtained, and all participants provided consent.

Reviewer #2: Appropriate methodology. Careful and well-prepared

Reviewer #3: The methods iusing human macrophages respond to infection by antimony-sensitive (zym 2.2) and antimony-resistant (zym 2.3) strains of Leishmania (Viannia) panamensis, using transcriptomic analysis in the presence or absence of pentavalent antimony (SbV). The methodology is clearly described and well aligned with the study’s objectives. Twelve clinical strains, six from each zymodeme were used, previously characterized by isoenzyme profiling and drug susceptibility. Ex vivo infections were performed using human macrophages derived from monocytes of 5 healthy donors. Analyses included RNA-seq, chemiluminescent quantification of reactive oxygen species (ROS), and robust bioinformatic tools (e.g., DESeq2, GSVA, GSEA).

**Results**

-Does the analysis presented match the analysis plan?

-Are the results clearly and completely presented?

-Are the figures (Tables, Images) of sufficient quality for clarity?

Reviewer #1: - The authors address the objectives stated in the introduction.

- Each figure addresses a different scientific question, and the accompanying narrative is convincing and easy to follow.

- The five figures take the reader through a clear journey, starting with looking at the main sources of variation (PCA plot; Fig 1), to effect of the two L. (V.) panamensis zymodemes (Fig 2), the added effect of antimony (Fig 3), the impact on global macrophage profile (Fig 4), and the functional effect on ROS production (Fig 5).

- The figures are easy to interpret, are provided in high resolution, are formatting is consistent. The figure provided in the Author Summary is very effective in condensing the take-home messages of the paper.

- The supplementary figures and tables serve an important role, helping the reader delve further into main causes of variation (S1 Fig) and providing the lists of enriched pathways (S3 and S4 Tables). S2 Fig is helpful in reiterating that zym 2.3 parasites show increased survival (as reflected by number of parasite transcripts) than zym 2.2 when infected macrophages are exposed to antimony.

Reviewer #2: The results adequately describe a natural resistance of a certain zymodeme of L. panamensis to antimonials.

Reviewer #3: The results are excellent and well presented; zym 2.3 strains induced broader transcriptomic modulation (646 vs. 268 genes), activated strong inflammatory pathways (chemokines, interferons), along with immunoregulatory axes (IDO1, IL4I1, *AHR), and were associated with reduced ROS production despite increased expression of antioxidant genes (e.g., SOD2) and reduced expression of antimony transporters (aquaporins and ABC family). In contrast, zym 2.2 strains elicited a proinflammatory profile involving NOTCH1 signaling and higher ROS output; when exposed to SbV, this inflammatory state persisted without induction of immunoregulatory mechanisms.

**Conclusions**

-Are the conclusions supported by the data presented?

-Are the limitations of analysis clearly described?

-Do the authors discuss how these data can be helpful to advance our understanding of the topic under study?

-Is public health relevance addressed?

Reviewer #1: - The authors convincingly argue the importance of elucidating not only differences between parasite strains, but also in how the immune system (in this case macrophages) responds to the different strains. Specifically in the case of antimony resistance, understanding differences in responses could help uncover the specific mechanisms of resistance.

- The main points presented in the discussion include the role of the IDO1/IL4I1-KYN-AHR pathways and ROS, macrophage polarisation, the effect of antimony, and the role of transporters. The statements made are all supported by the results, and they are discussed extensively and in the context of the literature.

- The limitations are described effectively and honestly, summarizing the potential issues with ex vivo models (e.g. simplification, lack of other cells, etc), as well as the need to validate results from transcriptomic analyses with protein-level or functional observations.

- The authors are effective in putting the results in the context of the disease. Antimony resistance is a widespread issue in the treatment of cutaneous leishmaniasis, and the authors' new insights add to our understanding of potential mechanisms of this resistance in L. (V.) panamensis.

Reviewer #2: The conclusions can be improved especially by associating these findings with clinical practice. In other words, to comment on whether or not this genetic marker should be performed before using antimony? What is the generation of knowledge to introduce or not in the treatment?

Reviewer #3: The conclusions support the hypothesis of the manuscript. The study is technically rigorous and offers novel insights, particularly in proposing that natural resistance to antimonial drugs is not limited to parasite biology but involves active manipulation of the host macrophage response a concept that is both original and biologically plausible, with important implications for the field of immunoparasitology and leishmaniasis therapy.

**Editorial and Data Presentation Modifications?**

Reviewer #1: Introduction

Line 30: The authors could consider using ‘in response to’ or similar, rather than ‘by’, as ‘by’ suggests that the response is by the parasites rather than the macrophages.

Methods

Line 168: It may be useful for the authors to provide details on conditions for the culture of L. (V.) promastigotes.

Results

Lines 242-248: There is discussion of 12 L. (V.) panamensis strains being used in Methods, while 6 are used for the RNA-seq (with macrophages from 4 different donors). Greater clarity could be provided on how the extra samples (6 parasites + 1 donor) were used to validate the reliability of results, and whether or not these were included in the final analyses (and if not, why not). The authors could also state how many repeats were used for each condition in the RNA-seq analyses - from the PCA plot in Fig 1B it appears like 11-12 for each infected experiment, and 4 for uninfected macrophages; this information could be included in the relevant figure legends.

Line 314: It is perhaps unclear what led the authors to look at SOD2 in the first place (and present it in Figure 2E), as it does not seem to feature in the top 10 genes or in the top pathways.

Lines 324-326: It would be helpful to guide the reader as to which data support this statement (e.g. Figure 2A or S3 Table, or both).

Lines 328-332: It is unclear which data support some of the genes/pathways mentioned here as downregulated. This is particularly the case for NOTCH1 signalling, which is not indicated as downregulated in S3 Table, but rather as upregulated with zym 2.2.

Lines 344-350: The interpretation of the UpSet plot in Figure 3E may be incorrect. For example, according to my interpretation, 260 is not the number of genes that are downregulated with antimony treatment of macrophages infected with zym 2.2, but rather it is the number of uniquely downregulated genes. The total number of downregulated genes with zym 2.2 would be the sum of all the bars in the UpSet plot that include it (260 + 131 + 104 +14 + 5 + 516 + 10 = 1040). Therefore, these few lines of text may require re-interpretation. The set sizes shown by the horizontal bars on the left of the UpSet plot would also be inaccurate if this is true (e.g. for zym 2.2 downregulated it shows ~500, when it seems the total should be 1040). In contrast, the set sizes in the UpSet plot in Fig 2D appear to be consistent with the sums of the values in the vertical bars.

Lines 386-387: ‘chemokine interactions’ do not appear to be highlighted in Gene Ontology biological processes, but rather in Reactome pathways (‘chemokine receptors bind chemokines’), so this may need to be modified.

Line 390: Which pathway(s) is specifically being referenced by ‘ABC transporter transcription activity’? Greater clarity would be helpful to identify how the authors arrived at their conclusions on the involvement of ABC transporters.

Discussion:

Line 450: The transporters are central to the message of the paper; however, they are mostly not presented in the figures/data, and are mainly just mentioned in the text (with the exception of one ABC gene in Fig 2A, and aquaporin pathways in S4 Table). The authors could consider presenting the differential expression of selected ABC and AQP genes in the form of a table or a graph as in Fig 2E, which could clearly show that these genes are significant.

Line 493: It may be worth reminding the reader that AHR was also upregulated in response to zym 2.3 (in addition to IDO and IL4I1).

Figures and Tables:

Line 885: in Figure 4B-C, it could be useful to have more decimal places for the FDR q values that are presented as 0.0, or otherwise present these values differently, to get a better idea of how they compare to the other values presented (e.g. 0.02 and 0.0004 in Figure 4A).

Line 947: In S3 and S4 Tables, p values are given to 2 d.p., so that many values are shown as 0.00, which may not be too helpful to the reader. The authors could consider presenting these values instead as a power of e or 10, so that it is possible to compare the significance of different pathways, rather than having many 0.00 values.

Line 947: It may also be useful for the authors to clarify that the tables in S3 and S4 show only (or up to) the top 10 significant pathways (e.g. by specifying in the table title/legends), as the ‘top 10’ aspect is only mentioned in the results when referring to S4 Table.

Reviewer #2: The conclusions can be improved especially by associating these findings with clinical practice. In other words, to comment on whether or not this genetic marker should be performed before using antimony? What is the generation of knowledge to introduce or not in the treatment?

Reviewer #3: However, some limitations should be considered.

(i) the ex vivo model does not fully replicate the in vivo immune environment, lacking adaptive components like T cells and cytokine.

(ii) functional validations for key genes such as IDO1 or transporters were not performed (e.g., through knockouts or pharmacologic inhibition).

(iii) the analysis was limited to a single time point, which precludes understanding of dynamic immune responses;

(iv) The results just showed on L. panamensis, which limits extrapolation to other Leishmania species.

I recommend acceptance with minor revisions, suggesting the inclusion of a more detailed discussion of these limitations, the proposal of functional validation in future studies, and an expanded reflection on the potential clinical applications of the findings, such as the use of AHR inhibitors as host-directed therapy.

**Summary and General Comments**

Reviewer #1: - In summary, this an exciting study which unveils differences in macrophage responses to different zymodemes of L. (V.) panamensis, which correlate with resistance to antimony. The results are very clearly presented and thoughtfully discussed.

- The authors (some) have previously published RNA-seq studies on L. (V.) panamensis infection, looking at the effect of different strains (self-healing vs chronic infection) on gene expression in both PBMCs and hMDMs (Gomez et al., 2021), as well as on neutrophils (Diaz-Varela et al., 2024). The authors (some) also investigated the transcriptomic responses of hMDMs to L. amazonensis and L. major (Fernandes et al., 2016). However, this study, to my knowledge, is the first extensive investigation of hMDM transcriptomic responses to L. (V.) panamensis, with the added dimension of exploring the role of antimony exposure on gene expression.

- The fact that the authors are able to extract important take-home messages from the almost infinite pool of RNA-seq data is a particular strength, and they are effectively able to focus on a small number of key messages: transporters, ROS, and macrophage polarisation.

- Another strength is that, despite being primarily an RNA-seq study, the data, results and conclusions are presented in a an accessible and clear manner, meaning that any scientist should be able to follow the narrative.

- The inclusion of functional analysis for ROS production is a valuable addition, and paves the way for future functional analyses to confirm the differences identified in this study at the transcriptional level.

Reviewer #2: Well-written manuscript with interesting findings. I suggest adding an association between the findings and clinical practice of treatment

Reviewer #3: In summary, this manuscript is a technically rigorous, scientifically novel, and conceptually impactful contribution to the field. It elucidates the role of host macrophage programming in parasite survival and antimonial resistance and provides a valuable framework for future investigations into host-directed strategies. I recommend acceptance with minor revisions aimed at strengthening the contextualization and translational framing of the findings. No ethical concerns or issues related to dual publication are apparent.

PLOS authors have the option to publish the peer review history of their article (what does this mean? ). If published, this will include your full peer review and any attached files.

**Do you want your identity to be public for this peer review?** For information about this choice, including consent withdrawal, please see our Privacy Policy .

Reviewer #1: No

Reviewer #2: **Yes: ** Valdir Sabbaga Amato

Reviewer #3: No

**Figure resubmission:****Reproducibility:** To enhance the reproducibility of your results, we recommend that authors of applicable studies deposit laboratory protocols in protocols.io, where a protocol can be assigned its own identifier (DOI) such that it can be cited independently in the future. Additionally, PLOS ONE offers an option to publish peer-reviewed clinical study protocols. Read more information on sharing protocols at https://plos.org/protocols?utm_medium=editorial-email&utm_source=authorletters&utm_campaign=protocols

---

## [Editor Report · Decision Letter 1]

24 Sep 2025

Dear Dr. El-Sayed,

We are pleased to inform you that your manuscript 'Interplay of human macrophage response and natural resistance of infection by L. (V.) panamensis to pentavalent antimony' has been provisionally accepted for publication in PLOS Neglected Tropical Diseases.

Best regards,

Ulisses Gazos Lopes

Academic Editor

Laura-Isobel McCall

Section Editor

Shaden Kamhawi

co-Editor-in-Chief

Paul Brindley

co-Editor-in-Chief

---

## [Editor Report · Acceptance letter]

Dear Dr. El-Sayed,

We are delighted to inform you that your manuscript, " Interplay of human macrophage response and natural resistance of infection by L. (V.) panamensis to pentavalent antimony ," has been formally accepted for publication in PLOS Neglected Tropical Diseases.

Best regards,

Shaden Kamhawi

co-Editor-in-Chief

Paul Brindley

co-Editor-in-Chief
